# Low-Dose Oral Iron Replacement Therapy Is Effective for Many Japanese Hemodialysis Patients: A Retrospective Observational Study

**DOI:** 10.3390/nu15010125

**Published:** 2022-12-27

**Authors:** Chie Ogawa, Ken Tsuchiya, Mineko Kanemitsu, Kunimi Maeda

**Affiliations:** 1Maeda Institute of Renal Research, Kawasaki 211-0063, Japan; 2Biomarker Society, Inc., Kawasaki 211-0063, Japan; 3Department of Blood Purification, Tokyo Women’s Medical University, Tokyo 162-8666, Japan

**Keywords:** oral iron therapy, hemodialysis, absolute iron deficiency, renal anemia

## Abstract

Western guidelines recommend the use of intravenous iron supplementation for hemodialysis patients. However, in Japanese patients with well-controlled inflammation, iron replacement may be achieved with oral iron supplementation. This study involved 108 courses in 77 outpatient hemodialysis patients who received low-dose oral iron replacement therapy. Data from baseline to week 28 of treatment were analyzed to identify factors associated with effectiveness. Changes over time in erythrocyte- and iron-related parameters and erythropoiesis-stimulating agent (ESA) dose were investigated in the effective group. A total of 84 courses (77.8%) satisfied the effectiveness criteria. Compared with the effective and ineffective groups, only C-reactive protein (CRP) was significantly different (*p* < 0.01). ROC curve analysis with efficacy as the endpoint showed a CRP cut point value of ≤0.1 mg/dL (area under the curve, 0.69; 95% confidence interval, 0.57–0.81). The relationship between serum ferritin and hemoglobin fluctuation by reducing the ESA dose showed a positive correlation (*p* < 0.001). In the ESA maintenance group, the serum ferritin gradually increased and then remained constant at about 60 ng/mL. Our data suggest that patients with CRP ≤ 0.1 mg/dL may benefit from low doses of oral iron supplementation. Approximately 60 ng/mL serum ferritin may be sufficient during stable hematopoiesis.

## 1. Introduction

Recent studies have found that transferrin saturation (TSAT), a measure of available iron, is associated not only with anemia control but also with prognosis and the development of cardiovascular disorders in patients with chronic kidney disease (CKD) [1,2]. This indicates the need for iron supplementation when iron deficiency is indicated as well as the importance of good iron metabolism that maintains available iron.

Although oral and intravenous iron replacement therapies are available, oral therapy is usually the first choice for treating absolute iron deficiency because oral preparations are safe and easy to administer via a physiological route [3]. However, patients on hemodialysis (HD) with chronic inflammation, which results in increased levels of hepcidin, have reduced iron absorption. For this reason, intravenous iron supplementation is recommended in Western guidelines [4,5]. However, the eligibility criteria for iron replacement therapy in HD patients remains controversial. The production of hepcidin, a regulator of iron metabolism, is enhanced by iron signaling. When iron is present in excess, hepcidin inhibits iron absorption from the intestinal tract and the use of iron in hematopoiesis [6].

In our study of the relationship between the concentrations of reticulocyte hemoglobin (CHr), s-ft, and hepcidin in HD patients, we found a positive correlation between s-ft and CHr when s-ft was <60 ng/mL, a flat correlation when s-ft was between 60 and 150 ng/mL, and a negative correlation when s-ft was >150 ng/mL [7]. These results suggest that s-ft < 60 ng/mL indicates an iron deficiency while s-ft > 150 ng/mL indicates iron excess, possibly resulting in impaired iron turnover due to increased hepcidin.

In the PIVOTAL trial, a recent randomized controlled trial (RCT), HD patients treated with higher doses of iron had a better prognosis, greater reduction in the dose of erythropoiesis-stimulating agents (ESAs), and no increase in infection risk compared with patients treated with lower doses [8,9]. However, there have been some studies where intravenous iron supplementation has been associated with an increased risk of oxidative stress-induced atherosclerosis [10,11] and infection [12,13]. Moreover, nonphysiological high-dose iron supplementation may exacerbate increased hepcidin, which in turn can exacerbate impaired iron metabolism.

Another study reported that oral iron supplementation is more likely than intravenous iron supplementation to maintain Hb levels and achieve TSAT ≥ 20% in HD patients [14]. Therefore, because of the higher retention of available iron and better safety profile, the use of oral iron supplementation may be preferable. Oral iron supplementation is known to influence gut microbiota. However, an oral iron dose of <50 mg/day is reported to reduce adverse events [15].

In the present study, we investigated the efficacy of low-dose oral replacement therapy in HD patients with s-ft < 60ng/mL, suggesting an iron deficiency, from our previous study. We also investigated changes in erythrocyte- and iron-related parameters over time in order to perform effective ESA as well as iron replacement therapy in maintaining Hb targets. Then, we speculated the cut-off value of s-ft, which is considered to be iron-sufficient in hematopoiesis.

## 2. Materials and Methods

### 2.1. Patients

This study involved 93 patients on outpatient hemodialysis at our hospital who received low-dose oral iron replacement therapy between January 2018 and April 2022. All patients received 3 sessions of 3 to 5 h of maintenance hemodialysis per week for at least 3 months.

All patients provided written informed consent permitting data sampling and analysis. The protocol for the study was approved by the ethics committee of the Biomarker Society, Inc., which comprises five committee members, including outside experts.

### 2.2. Methods

Iron replacement therapy was performed by administering sodium ferrous citrate at 50 mg/day or ferric citrate hydrate at 250 mg/day (60 mg of iron) for patients with s ft < 60 ng/mL. Anemia- and iron-related data during the 28-week period from the start of treatment were reviewed.

Blood samples were drawn at the first dialysis session each week. Anemia-related data were obtained by blood sampling twice a month. The dose of darbepoetin alfa (DA), long-acting ESAs, was adjusted to a target Hb level of 10–12 g/dL in accordance with the JSDT guidelines. At the discretion of the attending physician, iron supplementation was discontinued when s-ft > 100 ng/mL or Hb > 12 g/dL was achieved. s-ft, serum iron (Fe), total iron-binding capacity (TIBC), serum albumin (s-Alb), and C-reactive protein (CRP) were measured once a month. TSAT was calculated from Fe and TIBC (TSAT = Fe/TIBC × 100). Kt/V was measured using the single pool method, and the normalized protein catabolic rate was measured once a month. s-Alb was measured by the bromocresol green method. In cases where multiple iron doses were administered to the same patient, data from the start of each course of iron supplementation were used for analysis.

In order to verify the usefulness of oral iron supplementation, changes in erythrocyte- and iron-related factors for 2 months after the start of iron administration were compared with oral (oral group) and intravenous iron administration (IV group). In IV group, 40 mg of saccharated ferric oxide was administered once a week from weeks 2 to weeks 4, a total of 4 times, and blood sampling was performed on weeks 0, 5, and 9 (48 courses in 22 HD patients). An amount of 40 mg of saccharated ferric oxide once weekly is recommended as intravenous iron replacement therapy in Japanese HD patients by the JSDT guideline.

The criteria for the efficacy of iron administration were considered as follows: (1) Hb increased by ≥1 g/dL as proposed by Buttarello M, et al. [16] and Mitsuiki K, et al. [17], (2) Hb > 12 g/dL without an increase in DA, and (3) the DA dose decreased because DA was adjusted to maintain target Hb. To examine the effects of iron supplementation and DA on erythrocyte- and iron-related parameters, changes in those parameters over time were compared among the following three groups: patients who continued to receive iron for 28 weeks in the effective group and had their DA dose reduced (group 1) or maintained (group 2, n = 16) and those who discontinued iron before week 28 (group 3, n = 19). Then, to investigate the relationship between the fluctuation in red blood cell (RBC) count and DA dose, we examined the correlation of the difference between the highest and the week 27–28 values of RBC count [RBC (max-28wk)] with the difference between the DA dose at baseline and the lowest dose [DA (0wk-min)]. In addition, to investigate the relationship between s-ft and Hb fluctuation, the correlation of the highest s-ft value [s-ft (max)] with the difference between the highest and the week 27–28 values of Hb [Hb (max-28wk)] was also examined. Furthermore, we examined the baseline erythrocyte parameters associated with Hb > 12 g/dL in the effective group.

### 2.3. Statistical Analysis

Analyses were performed using SAS ver. 9.3 (SAS Institute, Cary, NC). Data are presented as the mean ± SD and the median with interquartile range. The *t*-test was used to compare groups in terms of normally distributed continuous variables, and the Mann–Whitney U test was used for other skewed continuous variables. The chi-square test was used to compare nominally scaled variables. The optimal cutoff values of CRP for the effect of iron administration and that of erythrocyte index at baseline for Hb > 12 g/dL within 12 weeks were determined by receiver operating characteristic (ROC) analysis using the Youden index. We used univariable logistic regression models to evaluate the impact of CRP on the effect of iron administration and that of Hb at baseline for Hb > 12 g/dL within 12 weeks. One-way repeated measures analysis of variance was performed for changes over time, and Bonferroni’s multiple comparison test was used for post hoc tests. Two-way repeated measures analysis of variance was used for comparison among groups 1–3 and CRP value over time between effective and ineffective groups. Five dates of CRP that showed temporary rise due to infection were excluded. To evaluate the relationship of RBC count with DA dose and that of serum ferritin with Hb, Pearson’s product–moment correlation coefficient and a generalized linear regression model were used. The values of DA dose that were not normally distributed were log-transformed before performing the above parametric analysis. Two-tailed *p*-values less than 0.05 were considered to indicate a statistically significant difference.

## 3. Results

### 3.1. Patients

A total of 128 courses of iron replacement therapy were administered to 93 patients with DA therapy (courses administered more than 3 months apart were counted separately). A total of 20 courses in 16 patients were excluded, including eight patients with increased ferric citrate hydrate dose due to high phosphorus levels, nine patients with a history of hospitalization before or after iron supplementation, two patients with hemorrhage, and one patient discontinued due to constipation.

Finally, data for 108 courses in 77 patients were analyzed. Mean age at the start of treatment was 68.7 ± 12.8 years and median duration of dialysis was 7.3 years. Of the 108 courses, 78 were in men and 40 were in patients with diabetes. The mean Hb was 10.5 ± 0.7 g/dL, TSAT 18.2 ± 5.4%, s-ft 25.8 ± 11.3 ng/mL, and median CRP 0.1 (0.05–0.25) mg/dL. Proton pump inhibitors (PPI) or histamine-2 (H2) blockers were used in 64 courses. The iron preparations were sodium ferrous citrate in 68 courses and ferric citrate hydrate in 40 courses (Table 1).

### 3.2. Compared between Oral and Intravenous Iron Replacement Therapy

Changes in Hb over time were significantly different between the two groups (*p* < 0.001, Figure 1a). Up to weeks 3–5, Hb increased in both groups, and there was no significant difference in change from baseline (*p* = 0.29). However, after 5 weeks without IV iron administration, a sustained increase in Hb was observed in the oral group but not in the IV group. As a result, changes in Hb at weeks 7–9 were significantly lower in the IV group (*p* < 0.001, Figure 1b). Changes in TSAT and s-ft over time were also significantly different between the two groups (*p* = 0.047 and <0.001, respectively, Figure 1c,d). Change from the baseline in TSAT was not significantly different between the two groups at weeks 3–5 and 7–9 (*p* = 0.15 and 0.21, respectively), but the TSAT value tended to be higher in the oral group during the observation period. Changes in s-ft were significantly higher at weeks 3–5 (*p* < 0.001) and lower at weeks 7–9 (*p* = 0.001) in the IV group.

### 3.3. Identification of Factors Associated with the Efficacy of Oral Iron Supplementation

Of the 108 courses, 84 (77.8%) satisfied the efficacy criteria.

Demographics, erythrocyte-, iron- and nutrition-related parameters, CRP, HD efficiency, antacid use, and iron preparations were compared with the effective and ineffective groups; only CRP was significantly different (Table 1, Figure 2a). ROC curve analysis with efficacy as the endpoint showed a cut point of CRP ≤ 0.1 mg/dL (sensitivity 63.1%, specificity 70.8%, area under the curve [AUC] 0.69, 95% confidence interval [CI] 0.57–0.81: Figure 2b). Logistic regression analysis showed a significantly lower odds ratio (OR) for patients with CRP > 0.1 mg/dL than for those with CRP ≤ 0.1 mg/dL (OR 0.24, 95% CI 0.09–0.65, *p* < 0.01: Table 2). In addition, CRP was significantly lower in the effective group than in the ineffective group during the observation period (*p* = 0.01; Figure 2c).

### 3.4. Erythrocyte- and Iron-Related Parameters and Change in DA Dose over Time

There were 14 patients in Group 1, 16 in Group 2, and 19 in Group 3 (excluding one patient who did not satisfy the discontinuation criterion). The average duration of iron replacement therapy in Group 3 was 18.3 ± 4.1 weeks. The time courses of Hb, RBC, and MCH were significantly different among the three groups (*p* < 0.001 for Hb and RBC; *p* = 0.03 for MCH). Hb and RBC were significantly elevated early during treatment in Groups 1 and 3 but began to decrease after DA dose reduction. In Group 3, a significant difference in RBC was observed at weeks 27–28. Group 2 showed a gradual increase in Hb and RBC with a significant increase in Hb after weeks 11–12. MCH was initially low, especially in Group 3, but gradually increased in all three groups and became significantly elevated after weeks 11–12 in all three groups. The values were similar at weeks 27–28 in all three groups.

The time courses of TSAT and s-ft were also significantly different among the three groups (*p* < 0.03 for TSAT; *p* < 0.001 for s-ft). The mean TSAT remained elevated compared with the baseline in all groups at above 20%. s-ft was also significantly elevated after weeks 3–4 in Groups 1 and 3 with the highest value reported in Group 3 among the three groups. In contrast, in Group 2, s-ft gradually increased and then remained constant at about 60 ng/mL (Figure 3) (Appendix A).

### 3.5. Relationships between RBC Fluctuation and DA Dose Reduction and between s-ft and Hb Fluctuation

Significant positive correlations were observed between RBC (max-28wk) and Log10 (DA) (0wk-min) (β-coefficient: 49.8 [95% CI: 33.5 to 66.1, *p* < 0.001]) as well as between s-ft (max) and Hb (max-28wk) (β-coefficient: 34.6 (95% CI: 27.2 to 40.2, *p* < 0.001; Figure 4).

### 3.6. Identification of Erythrocyte Parameters at the Start of Iron Supplementation That Was Associated with Hb > 12 g/dL

ROC curve analysis with Hb > 12 g/dL within 12 weeks as the endpoint showed cut points of Hb ≥ 10.7 g/dL (sensitivity 63.5%, specificity 82.1%, AUC 0.80, 95% CI 0.71–0.88), RBC ≥ 352 × 104/μL (sensitivity 63.5%, specificity 78.6%, AUC 0.703, 95% CI 0.60–0.81), and MCH ≥ 28.5 pg (sensitivity 23.1%, specificity 92.9%, AUC 0.53, 95% CI 0.42–0.65), with Hb being the most useful predictor (Figure 5). The subsequent logistic regression analysis showed a significantly higher OR for patients who had Hb ≥ 10.7 g/dL compared with those who had Hb < 10.7 g/dL (OR 8.0, 95% CI 3.3–19.4, *p* < 0.01; Table 3).

## 4. Discussion

In this study, low doses of iron replacement therapy showed an anemia-improving effect equivalent to that of the IV group, although it was a short-term comparison. In addition, TSAT values tended to be higher in the oral group than in the IV group.

About 80% of the courses in this study met the efficacy criteria, indicating that even low doses of iron replacement therapy are effective in many Japanese HD patients. When patient demographics, dialysis efficiency, nutritional status, antacid use, and type of iron preparations used were compared with the effective and ineffective groups, only CRP differed significantly between the two groups. It is well-known that inflammatory signals lead to increased levels of hepcidin, which inhibits iron absorption in the gastrointestinal tract [18], and the results of the present study confirm this as well. Furthermore, this study suggests that CRP > 0.1 mg/dL is associated with reduced efficacy of oral iron supplementation. CRP showed no significant changes during the observation period in both groups. Therefore, monitoring CRP before treatment initiation appeared to be useful for predicting the effect of oral iron therapy. Because there are few reports on the cutoff value of CRP for the efficacy of oral iron supplementation in HD patients, the present results may provide useful information for the selection of iron preparations.

We adjusted the ESA doses in accordance with the JSDT guidelines, with Hb targets of 10–12 g/dL, and reduced ESA doses when Hb > 12 g/dL. In the analysis of changes in erythrocyte- and iron-related parameters over time, RBCs in Groups 1 and 3 were significantly elevated in the first half of the observation period. Given the reported involvement of iron in the early differentiation of hematopoietic cells [19], iron deficiency may have suppressed the action of ESAs. Subsequently, in Group 3, the RBC count began to decrease after the ESA dose was substantially reduced and was significantly lower than the baseline value at weeks 27–28. Although iron supplementation was discontinued midway through the observation period, the mean s-ft increased rapidly to more than 100 ng/mL. The significant positive correlation between s-ft (max) and Hb (max-28wk) suggested that the rapid increase in s-ft was not caused by iron absorption but rather that the iron used for RBCs shifted to storage iron as the number of RBCs decreased.

In contrast, in Group 2, s-ft gradually increased and then remained constant at about 60 ng/mL. The y-intercept of the approximate straight line in the correlation plot between s-ft (max) and Hb (max-28wk) also indicates that s-ft would be maintained at around 60 ng/mL in the absence of Hb fluctuation. Eschbach et al. investigated iron absorption from the intestinal tract of both healthy volunteers and HD patients and found that the relationship between s-ft and iron absorption was nearly identical in both groups, where iron absorption decreased to almost 0% when s-ft reached about 60 ng/mL [20]. Another study in Japanese HD patients showed that Hb and s-ft were positively correlated in patients with s-ft < 50 ng/mL whereas Hb remained unchanged in patients with s-ft ≥ 50 ng/mL [21]. The present results also suggest that s-ft can be maintained at around 60 ng/mL with iron supplementation in patients with stable hematopoiesis.

There was a significant positive correlation between RBC (max-28wk) and DA (0 wk-min). Our data also suggested that a substantial reduction in DA dose is likely to lead to a decrease in RBC count, causing short-term iatrogenic Hb fluctuations. An association between Hb fluctuations and mortality risk has also been reported in HD patients [22], suggesting the importance of stable Hb management. In Group 2, Hb showed a gradually increasing trend, suggesting that Hb could be easily controlled. Group 3 showed that reducing the ESA dose is prone to result in Hb fluctuations, especially in the early phase of iron administration. So, we then examined baseline parameters associated with the achievement of Hb > 12 g/dL within 12 weeks. Because Hb is calculated by multiplying RBC and MCH, we examined these three parameters and found that Hb was the best predictor. The probability of achieving Hb > 12 g/dL within 12 weeks of iron supplementation was significantly higher in patients with Hb ≥ 10.7 g/dL at baseline. These results suggest that in patients with Hb ≥ 10.7 g/dL, a stable increase in Hb would be achieved by reducing the ESA dose to lower Hb followed by iron supplementation.

In the present study, the mean TSAT was maintained above 20% after iron supplementation. The percentage of patients with TSAT < 20% was 46.4% before the start of iron supplementation but decreased to a minimum of 3.6% during the course of treatment, although fluctuations were observed, suggesting that oral iron supplementation is also effective in maintaining TSAT.

At the KDIGO (Kidney Disease: Improving Global Outcomes) conference, some credence was given to the results of the PIVOTAL trial, but concerns over the lack of comparison versus oral supplementation and placebo were expressed and the need for further study was mentioned for the management of iron and anemia based on individual CKD patient characteristics and not only on population Hb and TSAT values [23]. Moreover, given the short median observation period in the PIVOTAL trial (2.1 years), long-term prognosis and iron accumulation are concerns in Japan where patients require HD for long periods of time [24]. Intravenous iron supplementation is reported to be associated with increases in intact fibroblast growth factor 23 level and suppression of the action of ESAs [25,26]. The DOPPS study showed that more Japanese patients had low CRP levels compared with their Western counterparts [27]. Optimal iron replacement therapy in Japan may differ from that in Western counterparts.

In the present study, oral iron supplementation was effective in many patients with s-ft < 60 ng/mL despite the low doses. It is thus important to select an appropriate iron preparation for each patient rather than simply resorting to intravenous iron supplementation.

This study had some limitations. Because this was a retrospective and observational study and the number of patients enrolled was small, there is a possibility that other confounding factors remain to be analyzed. Next, although a target Hb level was set, anemia therapy was administered at the discretion of each doctor. We expect to conduct a large-scale prospective study on the effects of oral iron therapy on Hb levels and iron status for HD patients in the future.

## 5. Conclusions

The results of this study suggest that in HD patients, effective iron replacement can be achieved even with a low dose of iron supplementation, especially in patients with CRP ≤ 0.1 mg/dL. The results also suggest that when starting iron replacement therapy for patients with Hb ≥ 10.7 g/dL, a stable increase in Hb is more likely to be achieved by reducing the ESA dose to lower Hb < 10.7 g/dL followed by iron supplementation. This study also showed that s-ft may reach a steady state around 60 ng/mL after iron supplementation. Therefore, approximately 60 ng/mL of s-ft may be the cutoff value for iron sufficiency in hematopoiesis.

## Figures and Tables

**Figure 1 nutrients-15-00125-f001:**
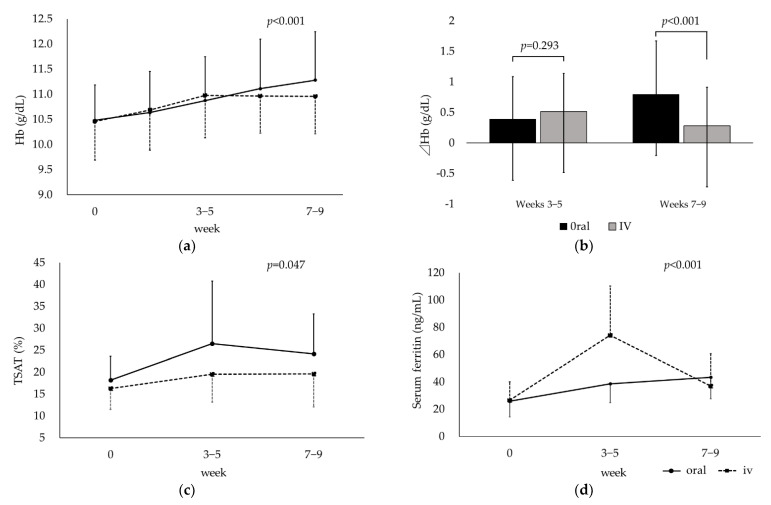
Comparison between oral and intravenous iron replacement therapy: (**a**) hemoglobin (Hb), (**b**) change in Hb from baseline, (**c**) transferrin saturation (TSAT), and (**d**) serum ferritin.

**Figure 2 nutrients-15-00125-f002:**
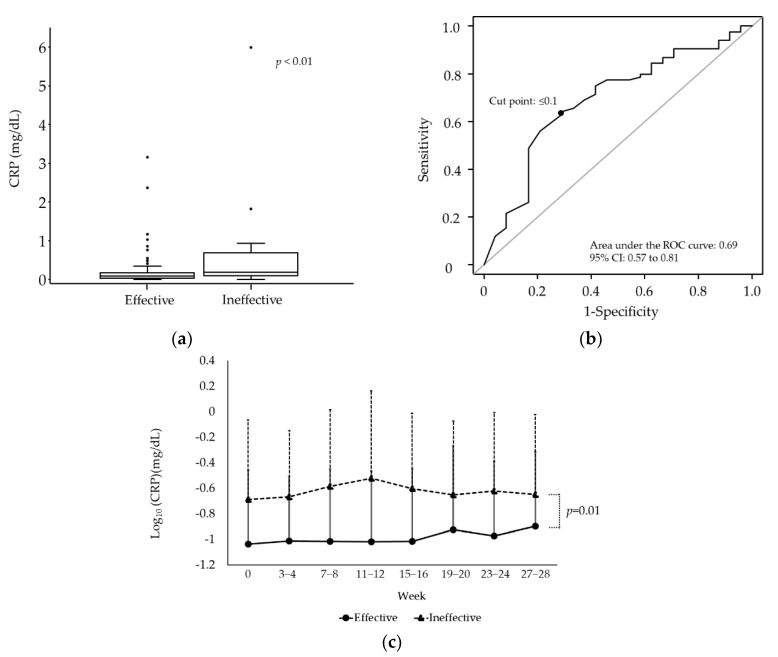
(**a**) Comparison of CRP levels between the effective and ineffective groups. The Mann–Whitney U test was used. (*p* < 0.01) (**b**) Receiver operating characteristic curves of CRP with effectiveness. (**c**) Changes in CRP over time.

**Figure 3 nutrients-15-00125-f003:**
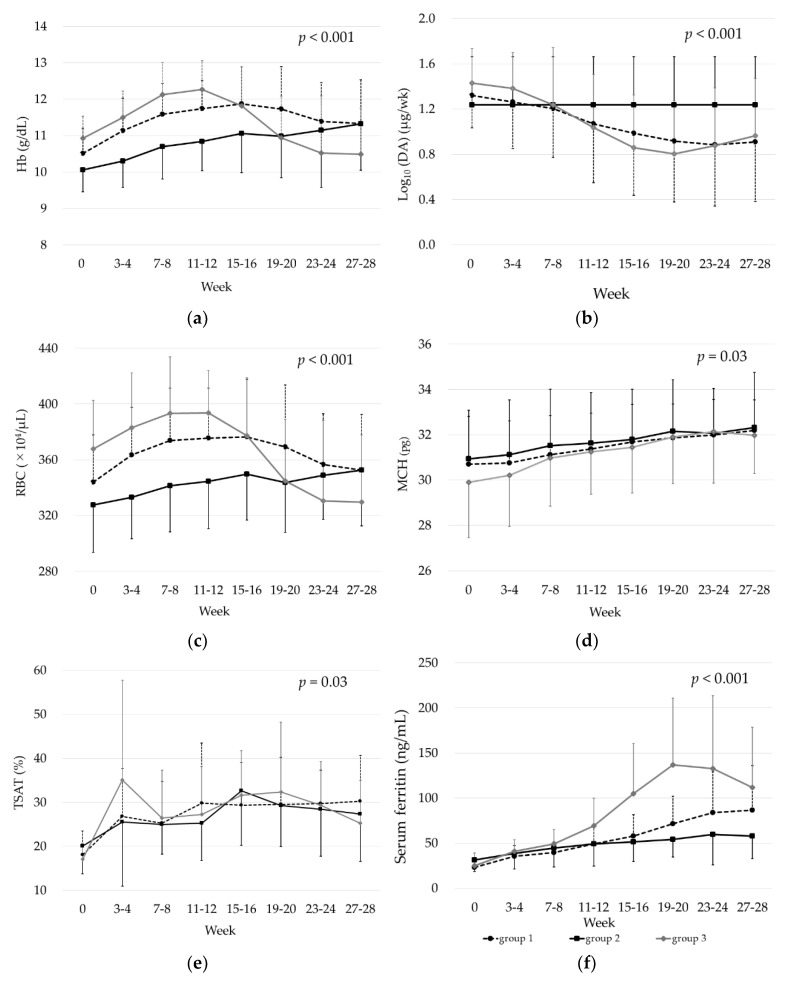
Two-way repeated-measures analysis of variance was used for comparison among Groups 1–3. (**a**) Hemoglobin (Hb), (**b**) Log10 (DA), (**c**) red blood cells (RBC), (**d**) mean corpuscular hemoglobin (MCH), (**e**) transferrin saturation (TSAT), (**f**) serum ferritin.

**Figure 4 nutrients-15-00125-f004:**
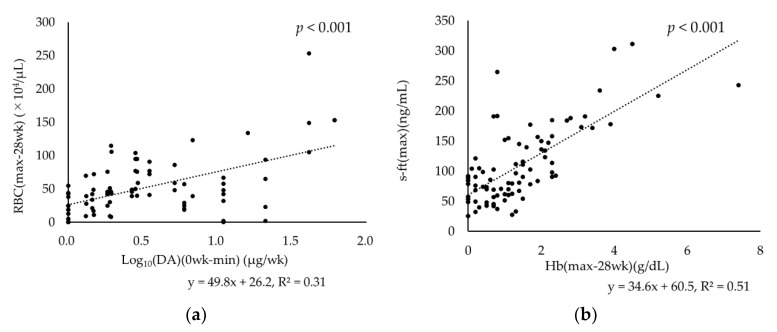
Relationship of (**a**) the change in RBC with DA dose and (**b**) serum ferritin levels with change in Hb levels.

**Figure 5 nutrients-15-00125-f005:**
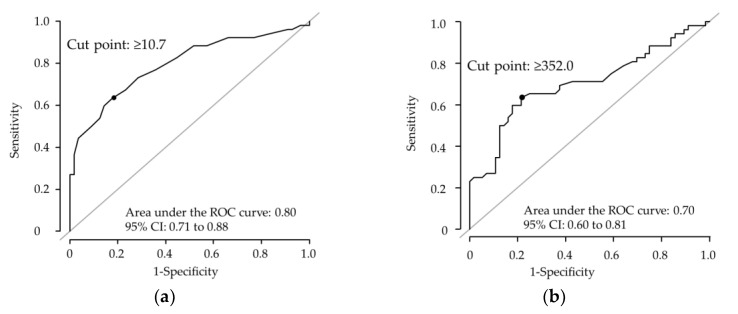
ROC curves of (**a**) Hb, (**b**) RBC, and (**c**) MCH with Hb > 12 g/dL within 12 weeks.

**Table 1 nutrients-15-00125-t001:** Patient characteristics.

Variables	All	Effective	No Effect	*p*-Value
N (courses)	108	84	24	
Age (years)	68.7 ± 12.8	69.0 ± 13.0	67.8 ± 12.9	0.67
Sex				
Men	78	60	18	0.80
Women	30	24	6	
Duration of dialysis (years) ^†^	7.3 (2.8–17.7)	6.7 (2.9–17.5)	8.0 (3.0–18.5)	0.89
Primary diagnosis				
Chronic glomerulonephritis	36	28	8	0.42
Diabetes nephropathy	40	30	10	
Renal sclerosis	24	18	6	
Other	8	8	0	
Hb (g/dL)	10.5 ± 0.7	10.5 ± 0.7	10.4 ± 0.7	0.28
RBC (×10^4^/μL)	345.9 ± 35.6	346.0 ± 36.7	345.5 ± 33.0	0.95
MCH (pg)	30.5 ± 2.1	30.6 ± 2.2	30.1 ± 1.3	0.41
MCV (fL)	96.3 ± 5.5	96.6 ± 5.8	95.3 ± 4.4	0.29
s-Fe (μg/dL)	51.4 ± 14.6	51.6 ± 14.8	50.4 ± 14.5	0.72
TIBC (μg/dL)	285.5 ± 41.2	284.4 ± 41.7	289.4 ± 41.0	0.60
TSAT (%)	18.2 ± 5.4	18.4 ± 5.6	17.7 ± 4.7	0.57
Serum ferritin (ng/mL)	25.8 ± 11.3	24.9 ± 11.8	28.9 ± 9.4	0.13
Albumin (g/dL)	3.4 ± 0.3	3.4 ± 0.3	3.5 ± 0.3	0.07
C-reactive protein (mg/dL) ^†^	0.10 (0.05–0.25)	0.09 (0.04–0.17)	0.19 (0.10–0.62)	<0.01
nPCR	0.92 ± 0.17	0.91 ± 0.14	0.94 ± 0.24	0.44
Kt/V	1.51 ± 0.25	1.52 ± 0.26	1.49 ± 0.23	0.64
Darbepoetin α (μg/week)	20 (10–30)	20 (10–32.5)	15 (10–22.5)	0.09
PPI/H_2_ blocker	64	52	12	0.35
Oral iron preparation				
Sodium ferrous citrate	68	55	13	0.34
Ferric citrate hydrate	40	29	11	

Values are shown as the number, mean ± standard deviation, or ^†^, median (interquartile range). Hb, hemoglobin; RBC, red blood cells; MCH, mean corpuscular hemoglobin; MCV, mean corpuscular volume; TIBC, total iron-binding capacity; TSAT, transferrin saturation; nPCR, normalized protein catabolic rate; Kt/V, urea removal status indicator; PPI, proton pomp inhibitor.

**Table 2 nutrients-15-00125-t002:** Univariable logistic regression models to evaluate the impact of CRP on the effect of oral iron therapy.

Variables	Courses, n	Effective, n	Odds Ratio	95% CI	*p*-Value
CRP (mg/dL)					
≤0.1	60	53	1.00		
>0.1	48	31	0.24	(0.09–0.659)	0.005

CI; confidence interval.

**Table 3 nutrients-15-00125-t003:** Univariable logistic regression models for the relationship between Hb levels at baseline and Hb > 12 g/dL within 12 weeks.

Variables	Courses, n	Effective, n	Odds Ratio	95% CI	*p*-Value
Hb (g/dL)					
<10.7	52	33	1.00		
≥10.7	32	29	7.99	(3.29–19.4)	<0.001

CI; confidence interval.

## Data Availability

Not applicable.

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
