# Peer review of "Low-Dose Oral Iron Replacement Therapy Is Effective for Many Japanese Hemodialysis Patients: A Retrospective Observational Study"

_nutrients, 2022, doi:10.3390/nu15010125_

Round 1
Reviewer 1 Report
The authors conducted a retrospective study to investigate the effectiveness of low dose oral iron replacement therapy for hemodialysis patients in Japan. I have a few comments that I hope the authors can address satisfactorily.
Major points
1. The authors showed that low dose PO iron is effective for their cohort of HD patients with 77.8% of treatment courses meeting the effectiveness criteria. Have the authors ever considered comparing the effectiveness of PO vs IV iron in their HD patients? The conclusion would have been convincing if the authors included those who received IV iron supplementation as a comparison arm.
2. In the introduction section, the authors should state clearly what their hypothesis is. Then in the discussion section, the authors should report whether they accept or reject their hypothesis based on their data/results.
Minor points
3. The introduction section seems a bit too long and unfocused. I suggest that the authors shorten the introduction and make it more focused.
4. Can the authors comment on how they determined the effectiveness criteria for iron supplementation (lines 107-108)? Please provide citations if the criteria were taken or adapted from prior literatures or guidelines.
Author Response
Reviewer 1
The authors conducted a retrospective study to investigate the effectiveness of low dose oral iron replacement therapy for hemodialysis patients in Japan. I have a few comments that I hope the authors can address satisfactorily.
Major points
- The authors showed that low dose PO iron is effective for their cohort of HD patients with 77.8% of treatment courses meeting the effectiveness criteria. Have the authors ever considered comparing the effectiveness of PO vs IV iron in their HD patients? The conclusion would have been convincing if the authors included those who received IV iron supplementation as a comparison arm.
Thank you for your suggestion.
According to the Japanese Society for Dialysis Therapy guidelines, intravenous iron replacement therapy should be 40 mg of saccharified ferric oxide once a week, up to a maximum of 13 courses. So, we cannot make comparison for 28 weeks. Therefore, we compared 2-month data in HD patients receiving oral and intravenous iron replacement therapy. Intravenous iron supplementation was administered from weeks 2 to 4 in 22 HD patients (48 courses). There was no difference in Hb elevation between the two groups for first month, suggesting that oral iron supplementation is as effective as intravenous iron supplementation. Interestingly, although Hb did not increase after discontinuation of intravenous iron in iv group, serum ferritin decreased. We did not mention this point in this article because it is different from the purpose of this study, but it may be necessary to further investigate iron dynamics after intravenous iron administration. Unfortunately, no data were available to match the reviewer's comments. Instead, we have added a drawing from the following data. If it does not answer the reviewer's intent, we will remove it. We would appreciate your kind attention.
We have added the sentences in “Methods”, “Results” and “Discussion” as follows, and also added Figure1.
“Methods”
In order to verify the usefulness of oral iron supplementation, changes in erythro-cyte- and iron-related factors for 2 months after the start of iron administration were compared between oral (oral group) and intravenous iron administration (iv group). In iv group, 40 mg of saccharated ferric oxide was administered once a week from weeks 2 to weeks 4, a total of 4 times, and blood sampling was performed on weeks 0, 5 and 9 (48 courses in 22 HD patients). 40 mg of saccharated ferric oxide once weekly is recom-mended as intravenous iron replacement therapy in Japanese HD patients by the JSDT guideline.(Line 95-102)
“Results”
3.2. Compared between oral and intravenous iron replacement therapy
Changes in Hb over time were significantly different between the two groups (P<0.001, Figure 1a). Up to weeks 3-5, Hb increased in both groups, and there was no significant difference in change from baseline (P=0.29). However, after weeks 5 without iv iron administration, a sustained increase in Hb was observed in the oral group, but not in the iv group. As a result, changes in Hb at weeks 7-9 were significantly lower in the iv group (P<0.001, Figure 1b). Changes in TSAT and s-ft over time were also significantly different between the two groups (P= 0.047 and <0.001, respectively, Figure 1c, d). Change from baseline in TSAT was no significant difference between the two groups at weeks 3-5 and 7-9 (P=0.15 and 0.21, respectively), but TSAT value tended to be higher in the oral group during the observation period. Changes in s-ft were significantly higher at weeks 3-5 (P<0.001) and lower at weeks 7-9 (P=0.001) in the iv group. (Line 160-171)
Figure 1. Compared between oral and intravenous iron replacement therapy
- a) Hemoglobin (Hb), b) Change in Hb from baseline, c) transferrin saturation (TSAT), d) serum ferritin.
“Discussion”
In this study, low doses of iron replacement therapy showed anemia improving- effect equivalent to that of the iv group, although it was a short-term comparison. In addition, TSAT values tended to be higher in the oral group than in the IV group. (Line236-238)
- In the introduction section, the authors should state clearly what their hypothesis is. Then in the discussion section, the authors should report whether they accept or reject their hypothesis based on their data/results.
Thank you for your suggestion.
We have rewritten introduction, discussion and conclusion section according to your suggestions.
“Introduction”
We added the sentences as follow.
In the present study, we investigated the efficacy of low-dose oral replacement therapy in HD patients with s-ft < 60ng/mL, suggested iron deficiency in our previous study. We also investigated changes in erythrocyte- and iron-related parameters over time in order to perform the effective ESAs as well as iron replacement therapy in maintaining Hb targets. Then, we speculated the cut-off value of s-ft, which is considered to be iron-sufficient in hematopoiesis. (Line 63-68).
“Discussion”
Some sentences have been deleted or moved to the introduction to make it easier to understand the answer to the hypothesis.
“Conclusion”
We added the sentences as follow.
This study also showed that s-ft may reach a steady state at around 60 ng/mL after iron supplementation. Therefore, approximately 60 ng/mL of s-ft may be the cutoff value for iron sufficiency in hematopoiesis. (Line 323-326)
Minor points
- The introduction section seems a bit too long and unfocused. I suggest that the authors shorten the introduction and make it more focused.
Thank you for your suggestion.
We have shortened the introduction to bring it into focus.
Can the authors comment on how they determined the effectiveness criteria for iron supplementation (lines 107-108)? Please provide citations if the criteria were taken or adapted from prior literatures or guidelines.
Thank you for your suggestion.
We have added the citations for determining the effectiveness criteria as follows.
“Methods”
The criteria for the efficacy of iron administration were considered as follows. (1) Hb increased by ≥1 g/dL proposed by Buttarello M. et al. [16] and Mitsuiki K. et al. [17], (2) Hb >12 g/dL without an increase in DA, and (3) the DA dose was decreased, because DA is adjusted to maintain target Hb. (Line103-105)

Reviewer 2 Report
The authors have documented that oral administration of iron is effective, provided that CRP values ​​are low. Given that CRP values ​​fluctuate over time and are variable, in the authors' opinion would it be necessary to monitor CRP in hemodialysis patients to prevent the ineffectiveness of oral iron therapy? please comment on
Author Response
Reviewer 2
The authors have documented that oral administration of iron is effective, provided that CRP values ​​are low. Given that CRP values ​​fluctuate over time and are variable, in the authors' opinion would it be necessary to monitor CRP in hemodialysis patients to prevent the ineffectiveness of oral iron therapy? please comment on
Thank you for your suggestion.
We have added figure of change in CRP over time, comment on it in the results (Figure 2c), analytical method in the statistical analysis and comment in discussion as follows.
“Statistical analysis”
Two-way repeated measures analysis of variance was used for comparison among groups 1–3 and CRP value over time between effective and ineffective groups. Five date of CRP that showed temporarily rise due to infection were excluded. (Line132-133)
“Results”
In addition, CRP was significantly lower in the effective group than the ineffective group during the observation period. (P=0.01; Figure 2c). (Line 184-185)
“Discussion”
There was no significant change in CRP over time from the start in both the effective group and the ineffective group. Therefore, monitoring of CRP before treatment initiation appeared to be useful for predicting the effect of oral iron therapy. (Line 247-249)

Round 2
Reviewer 1 Report
The authors have significantly revised their manuscript and adequately addressed my comments. I have no further comments.